# Evaluation of Porous Honeycomb-Shaped CuO/CeO$_2$ Catalyst in Vapour Phase Glycerol Reforming for Sustainable Hydrogen Production

Adrian Chun Minh Loy *, Shanthi Priya Samudrala and Sankar Bhattacharya *

Department of Chemical and Biological Engineering, Faculty of Engineering, Monash University, Clayton, VIC 3800, Australia
* Correspondence: adrian.loy@monash.edu (A.C.M.L.); sankar.bhattacharya@monash.edu (S.B.)

**Abstract:** This study presented an optimisation study of two-stage vapour-phase catalytic glycerol reforming (VPCGR) using response surface methodology (RSM) with a central composite experimental design (CCD) approach. Characterisation through Brunauer–Emmett–Teller analysis (BET), small-angle X-ray scattering (SAXS), scanning electron microscopy coupled with energy dispersive X-ray analysis (SEM-EDX), atomic force microscopy (AFM) and particle X-ray diffraction (PXRD) were carried out to understand the physiochemical activity of the honeycomb morphology CuO/CeO$_2$ catalyst. Notably, in this study, we achieved the desired result of glycerol conversion (94%) and H$_2$ production (81 vol.%) under the reaction condition of Cu species loading (10 wt.%), reaction temperature (823 K), WHSV (2 h$^{-1}$) and glycerol concentration (15 wt.%). From the RSM analysis, an optimum predicted model for VPCGR was obtained and further integrated into Microsoft Excel and Aspen Plus to perform an energy analysis of the VPCGR plant at a scale of 100 kg h$^{-1}$ of glycerol feed. As a whole, this study aimed to provide an overview of the technical operation and energy aspect for a sustainable frontier in glycerol reforming.

**Keywords:** Cu/CeO$_2$; aspen plus; energy analysis; glycerol reforming; heterogeneous catalyst

## 1. Introduction

Global warming and fossil fuel depletion are two of the greatest challenges of the humankind. Due to the acceleration of climate change's effect, the need for a paradigm shift from the use of conventional fossil fuels toward renewable energy in the global arena is becoming more important. The measures include developing energy legislation, supporting carbon credit programs and giving renewable energy tax credits and subsidies [1]. In this context, the urge for seeking an alternative renewable and environmentally friendly clean source, aligning to the 7th UN sustainable development goal "affordable and clean energy", is imperative to achieve a sustainable energy matrix [2–4].

Hydrogen (H$_2$) is one of the potential alternatives in reducing the society's reliance on fossil fuels, owing to its clean combustion characteristics and high calorific values [2,5]. Moreover, H$_2$ is a highly sought-after industrial commodity that can be utilised in applications ranging from power generation to chemical refineries [6,7]. Currently, most of the H$_2$ production is from unsustainable routes, more indicatively, from mature technologies such as hydrocarbon reforming and natural gas reforming [8]. Even though a huge amount of H$_2$ can be produced at a lower cost in these processes, the consumption of fossil resources is still contributing a huge amount of greenhouse gas emissions. Thus, developing a green, sustainable and efficient strategy for H$_2$ production is critical.

On the other hand, glycerol, one of the promising renewable top 12 building-block chemicals, has emerged as an attractive alternative for petroleum-based-product substitution [9,10]. By comparing with other renewable feedstock such as methanol and ethanol, glycerol (a) is less flammable and safer to handle due to a higher flashpoint (technical

standpoint), (b) is less hazardous and toxic (environmental standpoint), (c) is more thermo-dynamically favourable (energy standpoint) and (d) has higher $H_2$ production on the same basis of 1 mol of feedstock conversion [11–13]. Amongst all the possible routes of glycerol utilisation is the thermal transformation of glycerol to hydrogen via steam reforming (GSR). GSR is considered as one of the most economically feasible methods as compared with other thermal and biological counterparts [14,15]. From a circular carbon economy standpoint, GSR not only helps in lowering the production cost of biodiesel refinery but also plays an important role in the field of energy and fuels, which provides a blueprint for the development of the National Renewable Hydrogen Industry Plan and International Hydrogen Policy [16,17].

　　Based on the potential reactions of glycerol decomposition Equations (1)–(12) (see Table 1), GSR is highly favourable for $H_2$ production where 7 mol $H_2$ will be produced with the consumption of 1 mol glycerol molecule Equation (1). However, there are still many possible side reactions that might happen during thermal decomposition, including methanation, carbon deposition, glycerol oxidation and carbon gasification [18–20]. In order to eliminate the unfavourable side reactions, heterogeneous catalysts have been introduced into the reforming system such as catalysts based on Ni [21], Cu [22,23], Co [24,25], Pt [26,27] and Pd [28,29] to enhance the C–C and C–O cleavage. Amongst all the reforming's catalysts, the Cu-based catalyst can be considered as one of the cheapest with high effectiveness in breaking the C–O bonds of glycerol and inducing the water gas shift reaction due to the high polarisation Lewis acid active sites [30]. However, Cu metallic sites are prone to sintering and metal agglomeration at high temperatures (>723 K) and, consequently, cause the loss of active sites and catalyst deactivation [31,32]. To overcome these shortcomings, researchers tend to use high surface area, tunable physiochemical properties and high meta-stability materials as support to enhance its catalytic activity and lifespan (i.e., $Al_2O_3$, $SiO_2$ and $CeO_2$) [33,34]. To date, the literature reported on the influence of $CeO_2$ nanoparticle doping onto the Cu active sites is still limited, especially on engendering the glycerol reforming activity. For instance, in 2022, Wu et al. reported that the impregnation of Cu species onto the mesoporous $CeO_2$ can enhance the water gas shift reaction and inhibits methanation, leading to a high $H_2$ production; notably, the $H_2$ production rate has increased from 125.08 (Ni/$CeO_2$) to 195.57 $\mu$mol min$^{-1}$ g cat$^{-1}$ (Cu–Ni/$CeO_2$) [35]. Furthermore, the intrinsic oxygen vacancies or defect sites of the $CeO_2$ can induce the hydrogen spillover effect, which improves the glycerol conversion to $H_2$ production [36].

**Table 1.** Possible reactions in vapor phase glycerol reforming.

| | | |
|---|---|---|
| Glycerol Reforming | $C_3H_8O_3 + 3H_2O \leftrightarrow 3CO_2 + 7H_2$ | Equation (1) |
| Water gas shift reaction: | $CO + H_2O \leftrightarrow CO_2 + H_2$ | Equation (2) |
| Methanation: | $CO + 3H_2 \leftrightarrow CH_4 + H_2O$ | Equation (3) |
| | $CO_2 + 4H_2 \leftrightarrow CH_4 + 2H_2O$ | Equation (4) |
| Glycerol oxidation: | $C_3H_8O + 0.5O_2 \leftrightarrow 2CO_2 + 4H_2$ | Equation (5) |
| | $C_3H_8O_3 + O_2 \leftrightarrow CO + 2CO_2 + 4H_2$ | Equation (6) |
| | $C_3H_8O_3 + 1.5O_2 \leftrightarrow 3CO_2 + 4H_2$ | Equation (7) |
| | $C_3H_8O_3 + 3.5O_2 \leftrightarrow 3CO_2 + 4H_2O$ | Equation (8) |
| Decomposition of water | $2H_2O \leftrightarrow 2H_2 + O_2$ | Equation (9) |
| $CO_2$ reforming of methane | $CO_2 + CH_4 \leftrightarrow 2H_2 + 2CO$ | Equation (10) |
| Boudouard's reaction | $C + CO_2 \leftrightarrow 2CO$ | Equation (11) |
| Carbon gasification | $C + 2H_2O \leftrightarrow CO_2 + 2H_2$ | Equation (12) |

　　Response surface methodology (RSM) is a well-established mathematical and sta-tistical tool used to design empirical experimental models, build regression equations that describe the relative significance of each independent variable and determine the optimum reaction conditions based on the fit of the empirical model to a set of experimental data [37,38]. Under the RSM design-of-experiments central composite design (CCD) approach, a constant prediction of the variance at all points that are equidistant from the design centre can be attained, allowing a high precision of the predicted optimisation model.

With its inherent orthogonality feature, central composite designs can create orthogonal blocks, letting model terms and block effects be independently estimated and minimising the variation in the regression coefficients. Furthermore, its rotatable designs provide constant prediction of the variance at all points that are equidistant from the design centre, which further enhance the quality of predication [39]. Unlike a three-level full-factorial design, both linear and quadratic models are able to be determined. It can significantly reduce the number of experimental runs required and variability of the multifactor studies without compromising on the accuracy and reliability of the model experiments. Generally, a conventional factorial design approach is used for screening purposes instead of optimisation purposes via the linear model (less precise) [40]. On the contrary, the RSM-CCD approach is used as an optimisation tool to model the nonlinear relationship between the input factors and output responses. Over the years, many researchers have adopted this statistical tool in various fields, not limited to dry reforming [41], methanation [42], electrolysis [43] and hydrothermal liquefaction [44].

Herein, the present paper aimed to provide an insight on the vapour-phase catalytic glycerol reforming (VPCGR) using a highly porous honeycomb-shaped $CuO/CeO_2$ catalyst for sustainable $H_2$ production. The main objectives of this study were as follows: (a) developing an easy handling synthesis method for a highly porous $CuO/CeO_2$ catalyst; (b) analysing the textual structure properties of $CuO/CeO_2$ using various characterisation techniques (i.e., BET, SAXS, SEM-EDX, AFM, PXRD and Sy-PXRD); (c) performing an optimisation analysis to attain the highest glycerol conversion and $H_2$ production by varying the four different parameters (i.e., Cu loading, reaction temperature, weight hourly space velocity and glycerol concentration); and (d) evaluating the energy required for the VPCGR plant at a scale of 100 kg h$^{-1}$ of glycerol feedstock.

## 2. Results

### 2.1. Textual and Physicochemical Properties of the Catalyst

Table 2 shows the textual and physicochemical properties of the 10 wt.% $CuO/CeO_2$ nanoparticles (best case), including the actual Cu loading, crystalline size, surface area and height of the catalysts. Based on the SEM-EDX and XRF analyses, it is proven that the Cu particles were successfully impregnated on the surface of catalysts, forming a porous honeycomb-shaped morphology. In addition, from the SEM mapping analysis (Figure 1a), it can be seen that the Cu nanoparticles were homogeneously dispersed on the ceria support, indicating that the growth of the primary Cu nanoparticles was controlled and there was no agglomeration formation [45]. The observation is in good agreement with the SAXS (Figure 1b) and AFM (Figure 1c) analyses, corroborating that the average thickness (the lateral grain size) and particle size of $CuO/CeO_2$ were 17.50 and 25.00 nm, respectively, confirming the consistency of the homogeneous formation of CuO particulates on the surface of $CeO_2$.

**Table 2.** Textual and physicochemical properties of the catalyst.

| Catalyst | Cu Loading (wt.%) [a] | Cu$_x$O Loading [b] (wt%) | CuO (111) Crystalline Size (nm) | CeO$_2$ (111) Crystalline Size (nm) | Lattice Constant (Å) | Average Particle Size (nm) [f] | S$_{BET}$ (m$^2$g$^{-1}$) | Lateral Height of Catalyst (nm) [g] |
|---|---|---|---|---|---|---|---|---|
| 10 wt.%-CuO/CeO$_2$ | 8.52 | 11.12 | 35.6 [c], 39.2 [d], 42.1 [e] | 6.89 [c], 7.13 [d], 7.41[e] | 5.38 [c], 5.39 [d], 5.40 [e] | 17.50 | 14.28 | 25.00 |

[a] Determined using SEM-EDX analysis; [b] Determined using XRF analysis; [c] Determined using ex situ PXRD analysis; [d] Determined using in situ PXRD in $N_2$ atmosphere; [e] Determined using in situ synchrotron PXRD in $N_2$ atmosphere; [f] Determined using SAXS analysis; [g] Determined using AFM analysis.

Notably, via this impregnation-vacuum drying synthesis method, a small crystalline size of $CeO_2$ in the range of 6.89–7.41 nm with the lattice constant in between 5.38 and 5.40 Å assigned to the (111) plane of the cubic-phase structure of $CeO_2$ was attained. The $CeO_2$ crystalline size obtained in $CuO/CeO_2$ was much smaller as compared with other synthesis methods such as precipitation (9.5 nm) [46], incipient wetness impregnation

(16.0 nm) [47] and hydrothermal methods (29.6 nm) [48]. Moreover, highly significant CuO (111) monoclinic peaks at 36.5 and 38.7 were observed in synchrotron PXRD compared with that in in situ and ex situ lab-scale PXRD, providing better reliability and higher precision for the material insight towards nano- or sub-atomic-level catalytic mechanism [49].

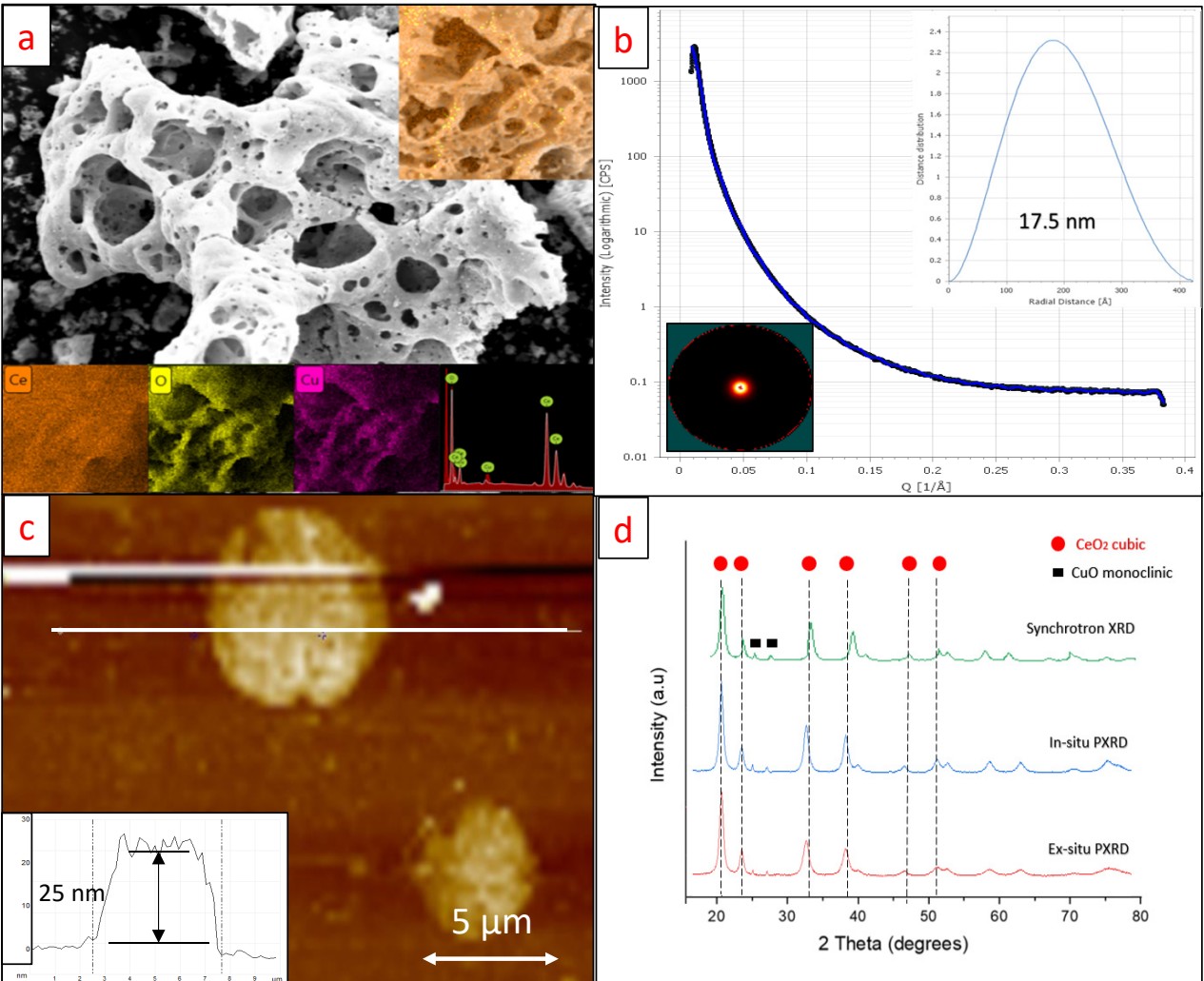

**Figure 1.** (**a**) Morphology of CuO/CeO$_2$ at 5000× magnification (insert: enlarged cross-section area at 20,000× magnification); (**b**) SAXS analysis: pair distance distribution function analysis; (**c**) lateral height analysis using AFM (insert: height and roughness of the catalyst); and (**d**) PXRD analysis of the catalyst.

### 2.2. Dual-Criteria Joint Optimisation Approach

The dual-criteria joint optimisation for VPCGR was based on the objective function of maximising the glycerol conversion (%) and hydrogen production (vol.%). The best-fit optimised model was chosen based on the highest $R^2_{adj}$ and $R^2_{pred}$, ensuring that the response variables are reliable and aligned with the model; obtaining the optimised model with the lowest standard deviation (SD), mean and $Ad_{eq}$ precision was also incorporated as an analysis criterion to avoid the issue of multicollinearity between predictors and responses [50].

#### 2.2.1. Mathematical Regression Model

The ANOVA regression responses on both rates of glycerol conversion (Y1) and hydrogen production (Y2) are shown in Tables 3 and 4, respectively. The significance of each regression coefficient was determined using a $p$-value, and notably, the $p$-value for

both models developed (<0.0001) was far less than 0.05, which indicates the significance of the model [51]. Apart from that, the regression analysis also shows that Y1 and Y2 were following different models, which are quadratic and linear, respectively. The equations of the significant terms obtained for the rate of glycerol conversion (Y1) and hydrogen production (Y2) are stated in Equations (13) and (14):

$$
\begin{aligned}
\text{Rate of glycerol conversion, } Y1 = {} & 81.39 + 2.54\,A + 8.88B + 2.12C + 1.95D - 0.5625AB \\
& - 1.56AC + 1.19AD + 0.8125BC - 0.9375BD - 0.4375CD + 0.6A^2 - 1.28B^2 - 1.4C^2 + 1.1\,D^2
\end{aligned}
\tag{13}
$$

$$
\text{Rate of hydrogen production, } Y2 = 70.78 + 1.54\,A + 3.88\,B + 1.79\,C + 0.5342\,D
\tag{14}
$$

**Table 3.** Analysis of variance (ANOVA) for response surface of quadratic model for Equation (14).

| Source | Sum of Squares | df | Mean Square | F-Value | *p*-Value |
|---|---|---|---|---|---|
| Model | 2097.56 | 14 | 149.83 | 38.88 | <0.0001 |
| A-Cu loading | 155.04 | 1 | 155.04 | 40.23 | <0.0001 |
| B-reaction temperature | 1488.37 | 1 | 1488.37 | 386.23 | <0.0001 |
| C-WHSV | 108.37 | 1 | 108.37 | 28.12 | <0.0001 |
| D-glycerol concentration | 71.14 | 1 | 71.14 | 18.46 | 0.0006 |
| AB | 5.06 | 1 | 5.06 | 1.31 | 0.2697 |
| AC | 39.06 | 1 | 39.06 | 10.14 | 0.0062 |
| AD | 22.56 | 1 | 22.56 | 5.85 | 0.0287 |
| BC | 10.56 | 1 | 10.56 | 2.74 | 0.1186 |
| BD | 14.06 | 1 | 14.06 | 3.65 | 0.0754 |
| CD | 3.06 | 1 | 3.06 | 0.7947 | 0.3868 |
| $A^2$ | 10.05 | 1 | 10.05 | 2.61 | 0.1271 |
| $B^2$ | 45.39 | 1 | 45.39 | 11.78 | 0.0037 |
| $C^2$ | 54.73 | 1 | 54.73 | 14.20 | 0.0019 |
| $D^2$ | 19.88 | 1 | 19.88 | 5.16 | 0.0383 |
| Residual | 57.80 | 15 | 3.85 | | |
| Lack of Fit | 46.47 | 10 | 4.65 | 2.05 | 0.2216 |
| Pure error | 11.33 | 5 | 2.27 | | |
| Cor total | 2155.37 | 29 | | | |
| Std. dev. | 1.96 | | $R^2$ | 0.9732 | |
| Mean | 80.57 | | Adjusted $R^2$ | 0.9482 | |
| C.V. % | 2.44 | | Predicted $R^2$ | 0.8594 | |
| | | | $Ad_{eq}$ precision | 23.6774 | |

**Table 4.** Analysis of variance (ANOVA) for response surface of linear model for Equation (15).

| Source | Sum of Squares | df | Mean Square | F-Value | *p*-Value |
|---|---|---|---|---|---|
| Model | 500.44 | 4 | 125.11 | 53.60 | <0.0001 |
| A-Cu loading | 57.04 | 1 | 57.04 | 24.44 | <0.0001 |
| B-reaction temperature | 360.37 | 1 | 360.37 | 154.38 | <0.0001 |
| C-WHSV | 77.04 | 1 | 77.04 | 33.00 | <0.0001 |
| D-glycerol concentration | 5.98 | 1 | 5.98 | 2.56 | 0.1220 |
| Residual | 58.36 | 25 | 2.33 | | |
| Lack of fit | 53.03 | 20 | 2.65 | 2.49 | 0.1586 |
| Pure error | 5.33 | 5 | 1.07 | | |
| Cor total | 558.80 | 29 | | | |
| Std. dev. | 1.53 | | $R^2$ | 0.8956 | |
| Mean | 70.80 | | Adjusted $R^2$ | 0.8789 | |
| C.V. % | 2.16 | | Predicted $R^2$ | 0.8454 | |
| | | | $Ad_{eq}$ precision | 24.8499 | |

On the other hand, the F-value Y1 (38.88) and Y2 (53.60) further confirmed the significance of both models where there was only a 0.01% chance that an F-value this large could occur due to noise [52]. From Figure S1, the values of $R_2$ and $Ad_{eq}$ precision for both

models were above 0.85 and 4, respectively, which implies that the model is reliable and can be used to precisely navigate the design space under adequate noise conditions [53]. Lastly, the coefficient of variation (C.V.), or known as the standard deviation divided by the mean obtained for both models, was less than 5, which signifies the accuracy of the models, in which there is a lower degree of variation to the mean value [54].

### 2.2.2. Parametric Analysis

The relationship of each independent variable was illustrated in contour plots as shown in Figure 2 (Y1) and Figure 3 (Y2). Each response surface was plotted against another response, which represents the combination of two reaction variables with another two-variable response fixed at a central level.

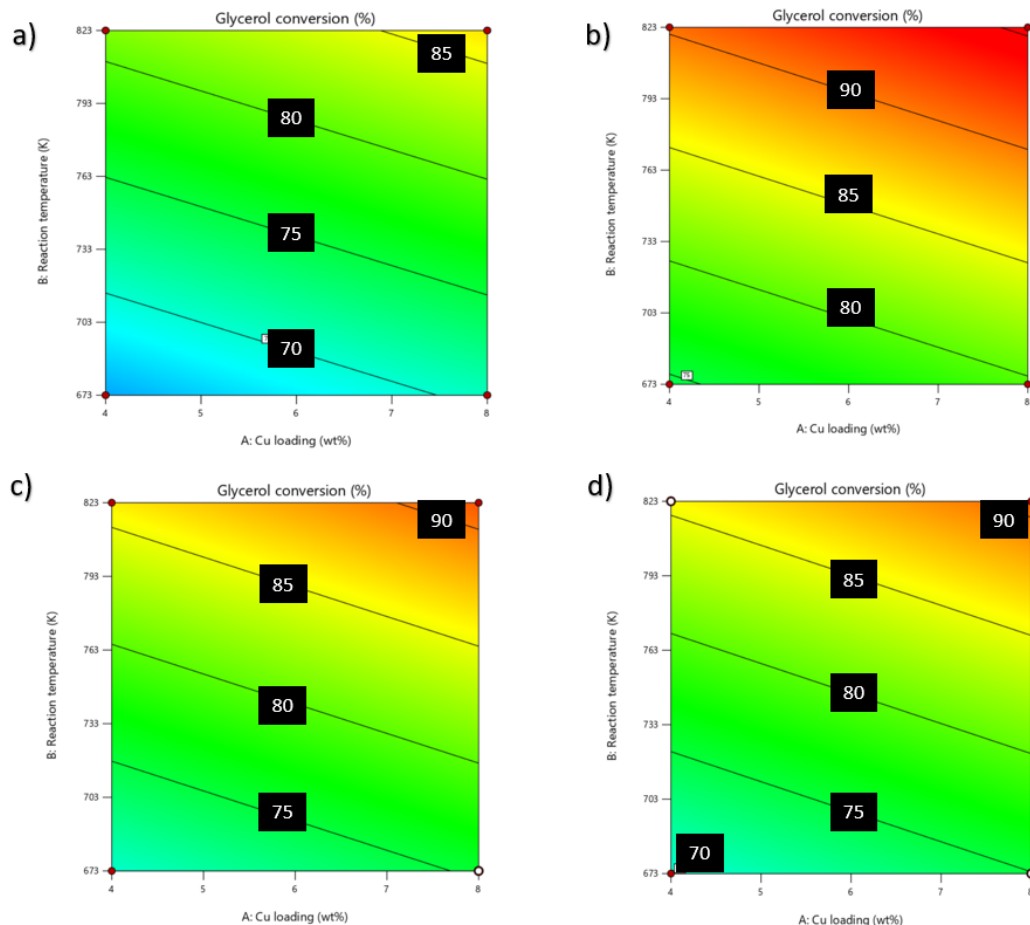

**Figure 2.** Counter plot for Y1 on reaction temperature vs. Cu loading: (**a**) WHSV and glycerol concentration are in minimum conditions; (**b**) WHSV and glycerol concentration are in maximum conditions; (**c**) WHSV is at minimum and glycerol concentration is at maximum; and (**d**) WHSV is at maximum and glycerol concentration is at minimum. (The black box indicates that the highest glycerol conversion can be obtained in the selected boundaries.)

### Effect of Cu Loading on CuO/CeO$_2$ Catalyst (wt.%)

Cu-based catalysts have been widely reported as effective active sites in bi- or multimetallic-site catalysts due to the superior performance for selective C–O bond cleavage as compared with its non-noble metal counterparts (i.e., Ni and Co), thus enhancing the selectivity towards H$_2$ production from glycerol [12,55]. Expectedly, the effect of Cu loading was directly proportional to the glycerol conversion, as there are more active sites available for C–C and C–O bond cleavage of glycerol molecules with an increase in Cu loadings. However, the Cu concentration on the CuO/CeO$_2$ catalyst was found to be not the most predominant

factor as compared with the reaction temperature in terms of glycerol conversion. On the other hand, remarkably, although with a high loading of Cu on the $CeO_2$ support (10 wt.%), no sign of a negative trend in the reduction of $H_2$ selectivity was observed. This phenomenon indicates that the prepared $CuO/CeO_2$ via this impregnation-vacuum drying synthesis method has suppressed the agglomeration of Cu clusters (i.e., inhomogeneity) during the nucleation process, which affects their catalytic cracking activity [56]. In all likelihood, a stronger interaction between Cu species and $CeO_2$ was developed by the impregnation-vacuum drying method, leading to better Cu species dispersion on $CeO_2$ [57].

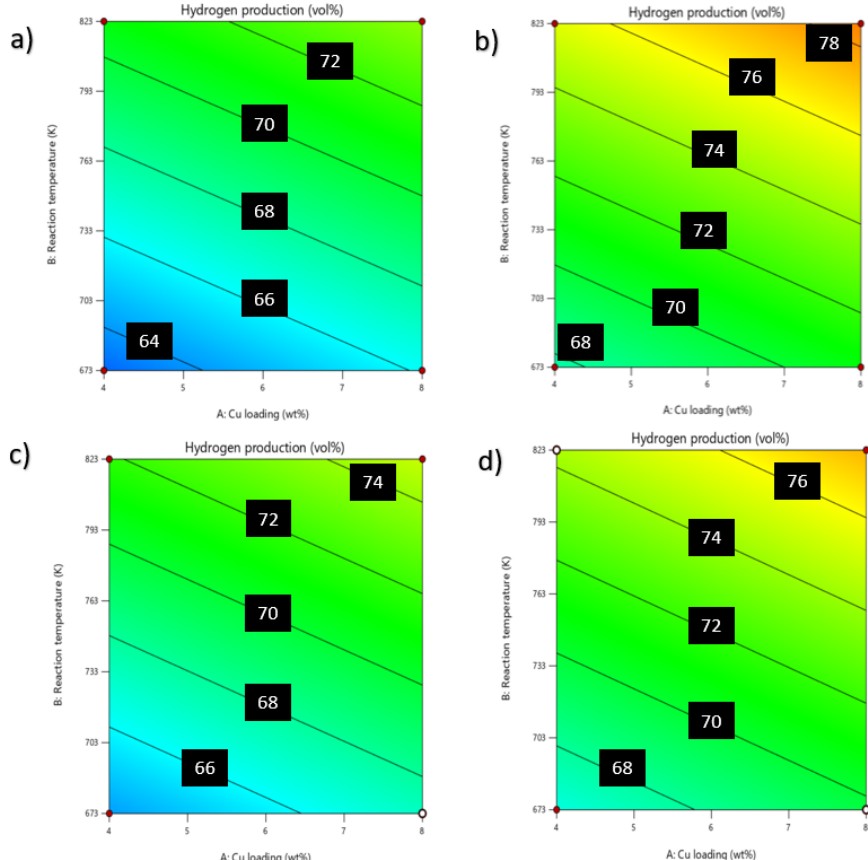

**Figure 3.** Counter plot for Y2 on reaction temperature vs. Cu loading: (**a**) WHSV and glycerol concentration are in minimum conditions; (**b**) WHSV and glycerol concentration are in maximum conditions; (**c**) WHSV is in minimum condition and glycerol concentration is in maximum condition; and (**d**) WHSV is in maximum condition and glycerol concentration is in minimum condition. (The black box indicates that the highest hydrogen production can be obtained in the selected boundaries.)

Effect of Reaction Temperature (K)

From Figures 2 and 3, it can be observed that temperature was the most dominant factor in enhancing the glycerol conversion. Even though in a low WHSV (i.e., 1 h$^{-1}$) and Cu loading (i.e., 4 wt.%) experimental condition, a positive effect in glycerol conversion can still be observed with an increase in the reaction temperature. This phenomenon can be explained through the Le Chatelier principle where an increase in the reaction temperature for an endothermic reversible reaction would favour the forward reaction; therefore, an increase in the reaction temperature above 773 K will significantly enhance both glycerol reforming Equation (1) and WGS Equation (2) reactions. Likewise, the elevation of the reaction temperature also shows a significant positive effect on the $H_2$ composition. From Figure 4, it can be seen that the production of $H_2$ was more significant as the reaction temperature increased above 723 K. For instance, the $H_2$ selectivity increased by 4 vol.% when the reaction temperature elevated from 623 to 723 K, whereas it increased by 7 vol.%

when the reaction temperature elevated from 723 to 823 K. This phenomenon is in good agreement with previous studies and might be due to the side reactions such as glycerol oxidation Equations (5)–(8) and $CO_2$ reforming of methane Equation (10), which further enhances the cracking of the volatile matter of glycerol to yield hydrogen [58].

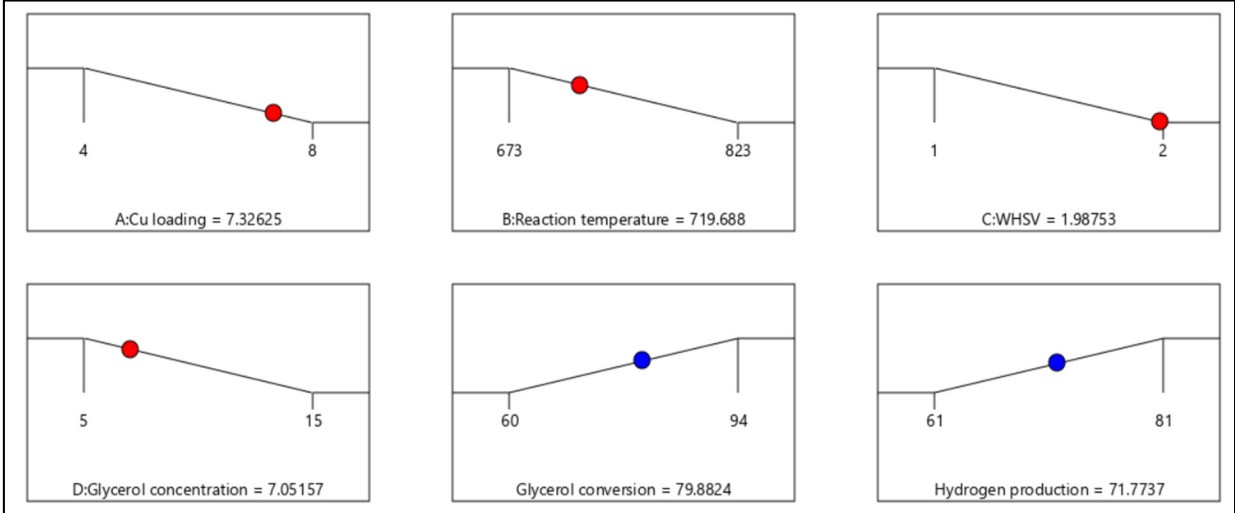

**Figure 4.** Predicted dual criteria optimisation results for the model (red: independent variable; blue: predicted results).

Effect of Weight Hourly Space Velocity ($h^{-1}$)

From the previous literature, an observation was reported in which the product gas composition, especially in glycerol reforming, was far from the equilibrium values, indicating that there is a need for an effective catalyst for the conversion of glycerol at a lower operating temperature [59,60]. On this basis, WHSV plays a crucial role in enhancing not only glycerol conversion but also $H_2$ selectivity. It can be seen from Figure 3a,b and Figure 4a,b that both $H_2$ selectivity and glycerol conversion rapidly increased with the increase in the WHSV (or known as the water–carbon ratio as the catalyst loading is fixed). However, when the WHSV was above 2 $h^{-1}$, the increase in $H_2$ selectivity and glycerol conversion becomes slower. This is because when the WHSV ratio was low (<1), the increase in the WHSV was favourable towards the glycerol reforming reaction. At the same phase, the increase in WHSV was also favourable towards the endothermic reactions such as WGS reaction Equation (2) and methane reforming reaction Equations (3) and (4). However, when the WHSV was above 2, most reactions almost reached equilibrium (a theoretical water–carbon ratio of 3 is the most optimum to complete the glycerol reforming reaction with water) [61]. On this basis, the optimum WHSV was found to be 2 $h^{-1}$; in future work, this number can be set as a reference for other researchers working in similar field.

Effect of Glycerol Concentration (wt.%)

Based on the plots in both Figures 2 and 3, it can be deduced that the effect of glycerol concentration on the glycerol conversion rate and $H_2$ production was less significant as compared with other variables. In addition, the increase in glycerol concentration will only enhance the $H_2$ production but will not be prominent towards the rate of glycerol conversion. Although at a high glycerol reaction condition of 20 wt. % at the reaction temperature of 748 K, Cu loading of 6 wt.% and WHSV of 1.5 $h^{-1}$ can yield 72 vol.% of $H_2$, glycerol conversion was below 85%, which is not feasible for industrial-scale glycerol reforming. Furthermore, much literature has reported that a high concentration of glycerol (>20 wt.%) will cause a decrease in the number of active sites of the catalyst due to the increase in surface coverage, thus hindering a steady flow mass transfer [26,62]. This is

because the increase in the viscosity of glycerol will reduce the mass transfer performance between the glycerol and liquid product on the catalyst surface (an immiscible three-phase system, which has a low mass transfer rate), thereby lowering the $H_2$ selectivity and glycerol conversion [63,64].

### 2.2.3. Post Confirmation Run Analysis

The confirmation test was conducted based on the predicted optimal reaction condition from the model (reaction condition: Cu loading = 7.3 wt%, reaction temperature = 719.7 K, WHSV = 1.99 $h^{-1}$ and glycerol concentration 7.05 vol%) (see Figure 4). Notably, based on the three-post-confirmation runs, very close results were attained under the optimal condition with a standard deviation of 3.38 and 1.53 for glycerol conversion and hydrogen production, respectively. Moreover, the 95% predicted intervals (i.e., both low and high 95% PI) were in a satisfying range, suggesting that this model is highly accurate, which can be used as a reference for future work (see Table 5).

**Table 5.** Confirmation Post Analysis of Experimental Work.

| Run | Glycerol Conversion (%) | Hydrogen Production (Vol%) | |
|---|---|---|---|
| 1 | 81 | 69 | |
| 2 | 79 | 70 | |
| 3 | 83 | 73 | |
| | Std Dev | 95% PI low | 95% PI high |
| Glycerol conversion (%) | 3.38 | 75.2 | 84.5 |
| Hydrogen production (vol%) | 1.53 | 69.6 | 73.8 |

### 2.3. Energy Analysis via IAMD Approach

The optimisation of the heat and energy integration networks of the VPGGR plant at a scale of 100 kg $h^{-1}$ of glycerol (reaction condition: Cu loading = 7.3 wt.%, reaction temperature = 719.7 K, WHSV = 1.99 $h^{-1}$ and glycerol concentration = 7.05 vol.%) was conducted using the Aspen Energy Analyzer. The heat and cooling utilities were found to be 158.2 MJ/h and −36.7 MJ/h, respectively, before implementing the heat exchanger network synthesis (HENS) modification. Notably, after the implementation of the heat recovery system, they were reduced to 111.3 MJ/h and −14.7 MJ/h for heat and cooling utilities, indicating that 68.9 MJ/h of energy reduction was achieved.

## 3. Materials and Methods

Figure 5 shows the research flow adopted for the work, starting from (a) catalyst synthesis, (b) design of experiment (DOE) for VPCGR, (c) response surface methodology optimisation, and (d) energy analysis via the integrated "Aspen Plus V12-Microsoft Excel-Design-Expert (IAMD) approach.

### 3.1. Samples and Catalyst Preparation

The analytical-grade pure glycerol (98%), cerium (III) nitrate hexahydrate and copper (II) acetate hydrate were obtained from Sigma Aldrich (St. Louis, MI, USA). The glycerol was used as a feedstock without further pretreatment, whereas the highly porous honeycomb-shaped $CuO/CeO_2$ was synthesised using a modified preparation method reported in previous studies [65,66]. Firstly, a mixture of 2.5 g cerium (iii) nitrate hexahydrate with 80 mL ethylene glycol and 15 mL of distilled water was placed in a beaker and stirred for 12 h at 353 K. After that, the slurry was vacuum-dried and calcined at 293 K and 673 K, respectively. To synthesise the honeycomb-shaped $CuO/CeO_2$, a predetermined Cu loading of between 2 and 0 wt.% was added into the aqueous dispersion solution of $CeO_2$

at a reaction temperature of 353 K for 12 h with continuous stirring at 600 rpm. Lastly, the remaining samples were centrifuged and washed with ethanol before annealing at 200 °C for 6 h in a 5 mol.% $H_2/N_2$ environment.

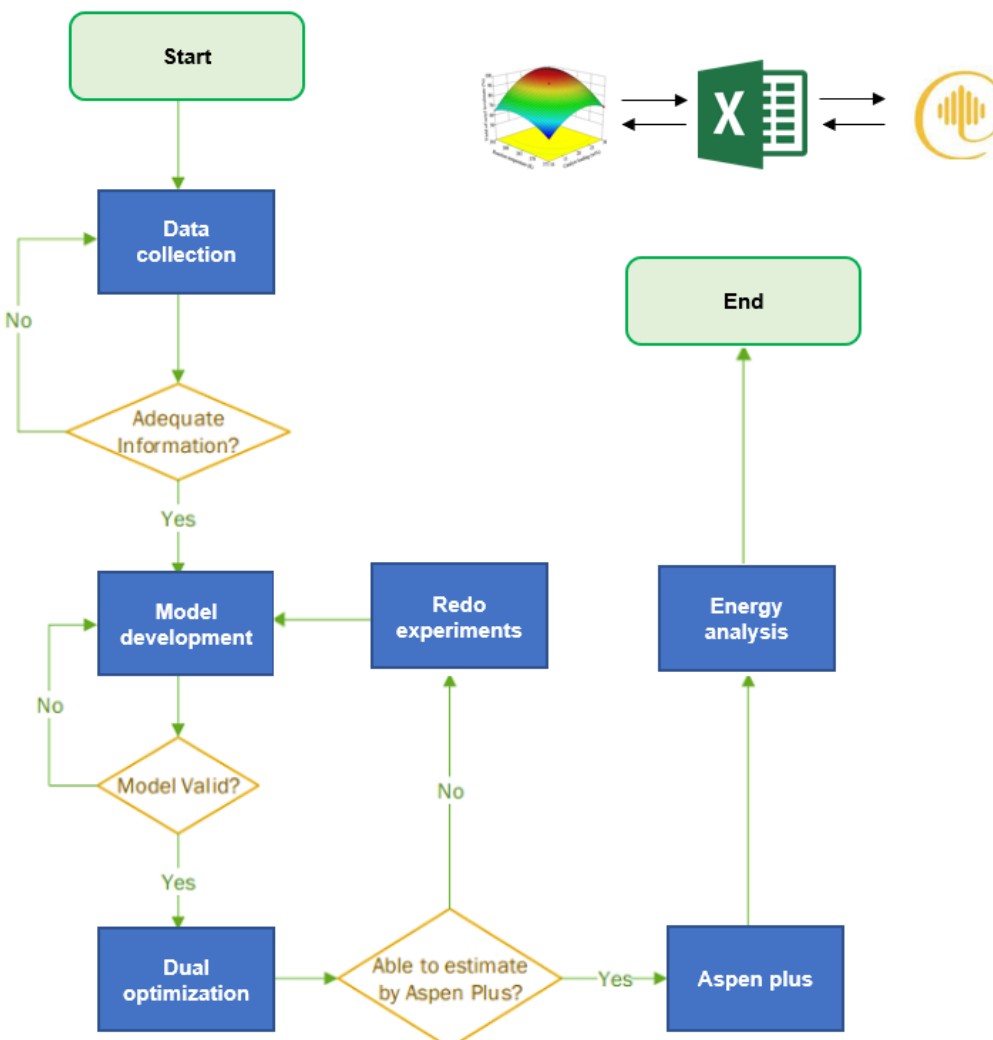

**Figure 5.** Research methodology flowchart of this study.

### 3.2. Catalyst Characterisation

A series of characterisation was performed to further elucidate the intrinsic physico-chemical properties of the highly porous honeycomb-shaped $CuO/CeO_2$, including scanning electron microscopy (SEM; Phenom XL Desktop SEM, Thermo Scientific, Waltham, MA, USA), small-angle X-ray scattering analysis (SAXS; N8 horizon, Bruker, Germany) and atomic force microscopy (AFM; Dimension Icon, Bruker, Germany); carried out using the AFM probe of diluted "$CuO/CeO_2$"/isopropanol by a simple dispersion method). Ex situ XRD and in situ PXRD (inert $N_2$ atmosphere) were performed using D8 Advance (Bruker, Germany), and synchrotron PXRD coupled with a diffraction beamline of 10BM-1 at a wavelength of 22 keV (0.56 Å) was used to analyse the crystallite phases of the samples (ANSTO, Sydney, Australia).

### 3.3. Vapour Phase Catalytic Glycerol Reforming

The VPCGR experiments were carried out in a fixed-bed-reactor system (see Figure S2). Firstly, a predetermined loading of the $CuO/CeO_2$ catalyst was placed in the middle of the fixed-bed reactor (Reactor:Shikoku Instrumentation μ Reactor EX, 2.45 GHz, maximum power 1 kW, length = 500 mm and internal diameter = 13 mm) in between the silica bed

materials. The catalysts were being reduced using 5 mol.% $H_2/N_2$ stream (50 mL/min) at 573 K for 3 h, followed by purging of pure $N_2$ at 50 mL/min to remove the remaining $H_2$ inside the system. Afterward, the vapour-phase glycerol (first stage—100 g of glycerol solution was heated above 523 K) was subjected into the fixed-bed reactor to induce a secondary cracking of the volatile matter of glycerol. Subsequently, the non-condensable gases were collected and analysed using gas chromatography utilising an online micro gas chromatograph (MicroGC 3000A) equipped with a TCD detector using a Molecular Sieve 5A column (10 m × 320 μm × 12 μm) with Ar as the carrier gas. Meanwhile, the glycerol concentration was measured using gas chromatography (GC 2010, Shimadzu, Australia) equipped with a flame ionisation detector (FID) and a column of Agilent DB-WAX, USA (Agilent, 30 m × 0.32; mm × 0.50 ID).

### 3.4. Optimisation Study

The optimisation of the VPCGR process was performed using a composite experimental design (CCD) response surface approach (Design-Expert® software version 13.0.1.0, State-Ease, Minneapolis, MN, USA) [52]. The independent variables (V), namely (a) the amount of Cu loaded on $CeO_2$ (2–10 wt.%), (b) reaction temperature (598–898 K), (c) weight hourly space velocity (0.5–2.5 $h^{-1}$), and (d) glycerol concentration (5–20 wt.%) were chosen to be optimised, and the response variables were (Y1) glycerol conversion (%) and (Y2) $H_2$ production (vol%) (see Table 6). To fit a second-order polynomial model with the experimental data through a response surface regression protocol, Equation (15) was adopted:

$$Y = \beta 0 + \sum j = 1n\beta jXj + \sum j = 1n\beta jjX2j + \sum j=1n - 1\sum I = 1n\beta jiXjXi \qquad (15)$$

where Y is the response variable (glycerol conversion or $H_2$ production); n denotes the number of analysed factors; Xi and Xj are the uncoded independent variables; and β0, βi, βii and βij are the model coefficients.

**Table 6.** Set of design of experiments based on CCD plot.

| Run | A: (Cu Loading, wt.%) | B: (Reaction Temperature, K) | C: (WHSV, $h^{-1}$) | D: (Glycerol Concentration, vol.%) | Y1: (Glycerol Conversion, %) | Y2:(Hydrogen Production, vol.%) |
|---|---|---|---|---|---|---|
| 1 | 4 | 673 | 1 | 15 | 68 | 63 |
| 2 | 6 | 598 | 1.5 | 10 | 62 | 61 |
| 3 | 2 | 748 | 1.5 | 10 | 81 | 67 |
| 4 | 4 | 823 | 1 | 5 | 79 | 71 |
| 5 | 8 | 823 | 2 | 5 | 90 | 79 |
| 6 | 6 | 748 | 1.5 | 10 | 82 | 70 |
| 7 | 10 | 748 | 1.5 | 10 | 87 | 73 |
| 8 | 8 | 673 | 1 | 15 | 84 | 69 |
| 9 | 8 | 673 | 2 | 15 | 81 | 68 |
| 10 | 8 | 823 | 1 | 5 | 88 | 72 |
| 11 | 6 | 898 | 1.5 | 10 | 93 | 78 |
| 12 | 4 | 823 | 2 | 15 | 88 | 75 |
| 13 | 8 | 673 | 1 | 5 | 73 | 66 |
| 14 | 4 | 673 | 2 | 5 | 72 | 68 |
| 15 | 4 | 823 | 2 | 5 | 91 | 74 |
| 16 | 10 | 823 | 2 | 15 | 94 | 81 |
| 17 | 6 | 748 | 1.5 | 10 | 82 | 71 |
| 18 | 4 | 673 | 2 | 15 | 72 | 69 |
| 19 | 6 | 748 | 1.5 | 20 | 80 | 73 |
| 20 | 6 | 748 | 1.5 | 5 | 81 | 69 |
| 21 | 6 | 748 | 1.5 | 10 | 79 | 72 |
| 22 | 6 | 748 | 1.5 | 10 | 80 | 73 |
| 23 | 6 | 748 | 1.5 | 10 | 82 | 72 |
| 24 | 4 | 823 | 1 | 15 | 84 | 70 |
| 25 | 6 | 748 | 2.5 | 10 | 81 | 72 |
| 26 | 6 | 748 | 1.5 | 10 | 83 | 72 |
| 27 | 4 | 673 | 1 | 5 | 64 | 64 |
| 28 | 6 | 748 | 0.5 | 10 | 71 | 68 |
| 29 | 8 | 673 | 2 | 5 | 72 | 70 |
| 30 | 8 | 823 | 1 | 15 | 90 | 74 |

*3.5. Energy Analysis*

To further investigate the feasibility of glycerol reforming in an industrial scale, energy assessment is an important criterion in decision-making, specifically to improve the facility energy performance [67]. In this study, the IAMD approach was adopted to assess the energy required from the VPGSR to provide a preliminary overview of VPGSR at a large industry scale. The VPCGR model is simulated using Aspen Plus V12, which comprises 6 units, namely mixer (B1), heat exchanger (H1, H2), reformer (R1, R2) and pump (P1) (see Figure S3). Firstly, the predicted experimental data obtained from the mathematical model were normalised in Microsoft Excel 2020 to obtain a complete 100% carbon balance (i.e., sum of $CO_2$, CO and $CH_4$). Then, the block named "VPCGR", known as the "black box model", was used in Aspen Plus V12 (NRTL-RK was used as the thermodynamic package) to link Microsoft Excel 2020. Lastly, to achieve an economically energy-saving plant, the Heat Exchanges Networks Synthesis (HENS) study was performed using the built-in Aspen Plus Energy Analyzer (Aspen Technology, Bedford, MA, USA). To follow the chemical industrial energy application boundary, 10 K was set as the minimum temperature difference $\Delta T_{min}$, ensuring an effective and feasible analysis.

**4. Conclusions**

In this work, we presented a new synthetic route to develop a honeycomb morphology $CuO/CeO_2$ catalyst with a highly effective catalytic reforming activity. Under the reaction conditions of Cu loading 8 wt.%, reaction temperature 823 K, WHSV 2hr$^{-1}$ and glycerol concentration 15 wt.%, the prepared $CuO/CeO_2$ managed to yield high glycerol concentration and $H_2$ production of 94% and 81 vol.%, respectively. Below are some of the key takeaways from this work:

(a) A small crystalline size of $CeO_2$ in the range of 6.89–7.41 nm was observed in PXRD, which is much lower compared with other synthesis methods, suggesting that the prepared impregnation-vacuum drying synthesis method is feasible to be adopted to a large-scale process.

(b) No Cu agglomeration was observed in the catalyst even at high Cu loading, suggesting that the Cu elements were homogeneously dispersed throughout the $CeO_2$ support.

(c) From the experimental work, the highest glycerol conversion (94%) and $H_2$ production (81 vol.%) can be obtained under the reaction conditions of the amount of Cu species loading (10 wt.%), reaction temperature (823 K), WHSV (2hr$^{-1}$) and glycerol concentration (15 wt.%).

(d) The predicted optimised condition with the lowest standard deviation (reaction condition: Cu loading = 7.3 wt%, reaction temperature = 719.7 K, WHSV = 1.99 hr$^{-1}$ and glycerol concentration 7.05 vol%) was deduced from the RSM CCD model, suggesting that this model can be applied in a large bench-scale study for benchmarking purposes.

(e) Principal component analysis (PCA) can be performed as a future work to identify the main correlations between the domain factors that enhance the $H_2$ production.

(f) Under the optimum condition, the HENS analysis shows that the VPCGR plant (scale: 100 kg hr$^{-1}$) requires 111.3 MJ/h and $-14.7$ MJ/h for heat and cooling utilities, respectively; this result can be set as a reference for scale-up purposes.

**Supplementary Materials:** The following supporting information can be downloaded at: https://www.mdpi.com/article/10.3390/catal12090941/s1, Figure S1: Predicted vs. actual data: (a) glycerol conversion and (b) hydrogen production. Figure S2: Schematic diagram of the VPCGR experimental set-up. Figure S3: Aspen Plus Flow diagram for VPCGR.

**Author Contributions:** A.C.M.L.: original draft, data curation, formal analysis, investigation and Software. S.P.S.: formal analysis and writing—review and editing. S.B.: formal analysis, supervision, writing—review and editing and investigation. All authors have read and agreed to the published version of the manuscript.

**Funding:** L.A.C.M. would like to thank the Australian Government for providing financial support (Australia Commonwealth Research Training Program) to this project. In addition, the authors would also like to acknowledge the Melbourne Centre for Nanofabrication (MCN) in the Victorian Node of the Australian National Fabrication Facility (ANFF) and Australian Nuclear Science and Technology (ANSTO) for AFM and synchrotron PXRD characterisations.

**Conflicts of Interest:** The authors declare no conflict of interest.

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
