# Peer review of "Evaluation of Porous Honeycomb-Shaped CuO/CeO2 Catalyst in Vapour Phase Glycerol Reforming for Sustainable Hydrogen Production"

_catalysts, doi:10.3390/catal12090941_

Round 1

Reviewer 1 Report

In this manuscript, authors reported some significant results on the study of vapor-phase glycerol reforming over Cu/CeO2 catalysts, and got optimized reaction conditions and optimum predicted model by using composite experimental design response surface approach. I recommend it can be accepted for publication after minor revision.

1)       About the particle size, authors mentioned that a small crystalline size of CeO2 in the range of 6.89-7.41 nm is attained. Concerning the fact that Cu particles are the main catalytically active sites for the reaction, I think it would be better to show more information about Cu particles, e.g., particle size and distribution states.

2)       It looks very strange to give an order about the effect of various factors in glycerol conversion and H2-production (i.e., reaction temperature, Cu loading and so on). I just feel it is no sense to make comparison between these factors.

3)       Some numbers or characters in figures are not very clear, e.g., in Fig. 1 and Graphic Abstract.

4)       There are some typos and grammar mistakes, for instances, “In 2022, Wu et.al. have reported the addition of Cu metallic sites to the mesoporous CeO2 to the Cu …...”; “This phenomenon indicates that the Cu/CeO2 catalytic preparation method……”; “the 95% predicted interval (low and high PI) are in a satisfying range”; “To synthesis the honeycomb Cu/CeO2”. Please polish the whole manuscript carefully.  

Reviewer 2 Report

The paper by Andrian et al. deals with a study of catalysts of copper dispersed on ceria employed vapour phase glycerol reforming for sustainable hydrogen production. Catalysts are characterized by different techniques in order to establish structure/activity relationships. The main conclusions emphasize the role of honeycomb morphology Cu/CeO2 catalyst has a small ceria crystallite size, and homogeneous copper distribution on ceria provides the higher activity. The subject is of particular importance from the practical point of view and toward the rational design of catalysts for nitrate reduction.  The manuscript is well written and interesting to read. I will recommend this manuscript for publication in Catalysts after minor revision.

Specific comments are as follows.

  1. According to the XRD results, the Cu/CeO2 catalyst mainly contained CuO species. It is better to name the catalysts as CuO/CeO2.
  2.  On page 4, tiny labels in Figure 1b are difficult to read.
  3. The authors should include the sample preparation for AFM analysis in the experimental details.

Reviewer 3 Report

This paper proposes the optimization of two-stage gas phase catalytic glycerol reforming (VPCGR) using the response surface method (RSM) and the central compound empirical psychological design (CCD) method. Various characterization methods are introduced to understand the physical and chemical activity of honeycomb Cu/CeO2 catalysts. This article outlines the sustainable cutting-edge technological operation and energy aspects of glycerol, which is an interesting topic. There are also some problems in the manuscript. The specific modifications mainly include the following:

1. In the introduction, there is only a few introduction to why glycerol is used. Please introduce the advantages of glycerol over other substances like methanol and ethanol. The hydrogen production from stem reforming of ethanol (e.g. ACS Catalysis 2013, 3, 5, 975–984) and methanol (e.g. Catalysts 2022, 12(7), 747) should also be cited.

2. In the introduction, please introduce in detail the examples used by RSM and CCD methods for specific research, and write down what good results the two methods have achieved, so as to reflect the advantages of these two methods and why they should be used.

3. Table 1 lists a Cu/CeO2 catalyst with ~10% Cu. We noted that the authors have also prepared other Cu/CeO2 catalysts with other Cu loading. How did the copper loading affect the catalyst?

4. What do the numbers in the black boxes in Figures 2 and 3 represent? Please explain them in the diagrams of each figure. In addition, the figures are not clear, especially the name and date of the coordinate axis are very blurry.

5. When discussing the effect of glycerol on the reaction, are there any specific examples to prove that glycerol has no effect on the reaction product.

6. There are many typo mistakes. For example, some references have incorrect formats, and some chemical formulas have no subscripts.

7. All research conclusions should be included in the summary. Please supplement it.

8. The glycerol conversion, hydrogen production/selectivity and carbon balance should be provided.

9. The reaction temperature is very high (823K) for glycerol conversion. How could the authors avoid the side reactions like RWGS reaction?

10. The size of CeO2 is about 6.89 nm from the XRD results. The other characterizations like TEM should be done to confirm the CeO2 size.  

Round 2

Reviewer 3 Report

The authors have revised carefully. It can be accepted now.